# Photodynamic Inactivation of Human Herpes Virus In Vitro with Ga(III) and Zn(II) Phthalocyanines

**DOI:** 10.3390/v16121937

**Published:** 2024-12-18

**Authors:** Neli Vilhelmova-Ilieva, Vanya Mantareva, Diana Braikova, Ivan Iliev

**Affiliations:** 1Department of Virology, Stephan Angeloff Institute of Microbiology, Bulgarian Academy of Sciences, 26 Georgi Bonchev, 1113 Sofia, Bulgaria; 2Institute of Organic Chemistry with Centre of Phytochemistry, Bulgarian Academy of Sciences, Bld. 9, 1113 Sofia, Bulgaria; mantareva@yahoo.com (V.M.); diana.braykova@orgchm.bas.bg (D.B.); 3Institute of Experimental Morphology, Pathology and Anthropology with Museum, Bulgarian Academy of Sciences, 1113 Sofia, Bulgaria; taparsky@abv.bg; 4Department of Biotechnology, University of Chemical Technology and Metallurgy, 8 Kliment Ohridski, 1756 Sofia, Bulgaria

**Keywords:** herpes simplex virus type 1, viral replication, virucidal effect, viral adsorption, enveloped viruses, phthalocyanines, photodynamic inactivation

## Abstract

Photodynamic inactivation (PDI) has been revealed as a valuable approach against viral infections because of the fast therapeutic effect and low possibility of resistance development. The photodynamic inhibition of the infectivity of human herpes simplex virus type 1 (HSV-1) strain Victoria at different stages of its reproduction was studied. PDI activity was determined on extracellular virions, on the stage of their adsorption to the Madin-Darby bovine kidney (MDBK) cell line and inhibition of the viral replication stage by application of two tetra-methylpyridiloxy substituted gallium and zinc phthalocyanines (ZnPcMe and GaPcMe) upon 660 nm light exposure with a light-emitting diode (LED 660 nm). The PDI effect was evaluated on extracellular virions and virus adsorption by the terminal dilution method and the change in viral infectivity, which was compared to the untreated control group. The decrease in viral titer (Δlgs) was determined. The effect on the replicative cycle of the virus was determined using the cytopathic effect inhibition (CPE) assay. The direct influence on the virions showed a remarkable effect with a decrease in the viral titer more than 4 (Δlg > 4). The influence of the virus to the cell on the stage of adsorption was also significantly affected by the exposure time and the concentration of applied photosensitizers. A distinct inhibition was evaluated for ZnPcMe at the viral replication stage, which demonstrated a high photoinactivation index (PII = 33.0). This study suggested the high efficacy of PDI with phthalocyanines on HSV-1 virus, with full inhibition caused by the mechanism of singlet oxygen generation. These promising data are a good basis for further investigations on the PDI application against pathogenic viruses.

## 1. Introduction

One of the most common human pathogens is herpes simplex virus type 1 (HSV-1), which is mainly associated with disease of the area around the mouth but is also increasingly registered with genital appearances. Herpetic keratitis is particularly severe, and potentially fatal diseases include encephalitis in newborns and disseminated infections in patients with defects in cellular immunity [1,2,3,4]. According to the World Health Organization, approximately 3.8 billion people under the age of 50 (64.2%) worldwide have HSV-1 infection [5]. Herpes simplex virus type 1 (HSV-1) is a member of the Alpha-herpesviridae subfamily. Its structure contains linear dsDNA, an icosahedral capsid, and an outer envelope. After primary infection, a latent state of the virus is established in the dorsal root ganglia and the trigeminal ganglia. Certain stressors such as illness or fever, sun exposure, menstrual period, injury, emotional stress, and surgery can reactivate the virus and lead to the development of recurrent infection [5]. During the latent period, lytic viral transcripts/proteins are not expressed except for the latency-associated transcript (LAT) [6,7].

Three classes of drugs are clinically approved for the treatment of HSV infections, and all of them are developed to block the replication of viral DNA. These are acyclic guanosine analogs [8,9,10,11,12], acyclic nucleotide analogs [13,14], and pyrophosphate analogs [13,14]. The most widely used is acyclovir-ACV with active compound 9-(2-hydroxyethoxymethyl) guanine. A series of anti-HSV nucleoside analogs of ACV, such as valacyclovir, famciclovir, and ganciclovir, are also commonly used in therapeutic practice [4,15]. The current therapeutics can reduce the duration and severity but cannot cure the infection itself. The studies showed that a long-term treatment with ACV and its derivatives can lead to drug resistance [16,17,18,19,20,21]. This becomes an important clinical problem, especially in immunocompromised patients [4,15,17]. Therefore, there is a need to develop new, more effective, low-toxicity anti-HSV drugs with different mechanisms of action, including the antiviral targets and the mechanisms involving new antiviral molecules [4,15]. This option is also the strategy of the World Health Organization in the prevention and control of HSV infections [5].

The last global pandemic crisis reinforced the research and development of novel photosensitizers as antiviral drugs, and photodynamic inactivation (PDI) appears a therapeutic method for the monitoring and prompt inactivation of the actual viruses [22,23,24]. PDI has been evaluated with a promising potential, especially for emergency cases without alternatives [25]. The photodynamic action was firstly observed on pathogenic bacteria, but lately, it has been clinically accepted for anticancer therapy with porphyrin derivatives [26]. Recently, PDI was reconsidered for application against viruses and virus-induced acute illnesses because of the fast growth of the resistance and the danger of appearance of the future epidemic circumstances [27]. The photodynamic procedure includes a photosensitizer (PS) with absorption in visible or near-infrared spectra, a light source with a proper spectrum of irradiation, and optimal light parameters, as well as an oxygen surrounding [28]. The PS application and its light activation in the presence of oxygen initiates the formation of triplet excited state PS molecules with the capability to react with the surrounding oxygen. The process ends with the production of reactive oxygen species such as singlet oxygen and superoxide ion and hydrogen peroxide and other very reactive species with a high oxidation potential. They may damage the structures of the viral envelope, such as lipids and proteins. Changes can happen in the structure of the viral nucleic acid, and all these damages can lead to stopping viral infection and the formation of new viral particles. Many scientific efforts have been made in the development of more effective photoactive compounds, including the derivatization of known photoactive compounds, which aims to design the PS with the desired assets for antivirals [29,30]. A recent study suggested that the physicochemical properties of PS and the virus compartment’s envelope, capsid, and genome may have a significant impact on the interactions between PS and virus and influence on the efficiency of the PDI outcome [31].

The promising heterocyclic photosensitive compounds are phthalocyanines (MPcs, M: metal ion) with promising potential for the inactivation of viruses. Presently, they are in usage for the disinfection of blood products with aluminum complexes [32]. A great number of MPcs have been synthesized and studied against viruses, among which are VSV, HSV, and HIV [33,34,35]. The effect was found only on enveloped viruses, and almost no effect was persuaded on the non-enveloped viruses [36,37].

In the present study, the antiviral effect on HSV-1 upon the photodynamic action of two phthalocyanine complexes (ZnPcMe and GaPcMe, Figure 1) at LED 660 nm exposure was evaluated. The photodynamic activity was determined at different stages of viral reproduction, such as the influence on extracellular HSV-1 virions, the stage of adsorption of virus particles to host cells, and the intracellular replicative cycle of HSV-1.

## 2. Materials and Methods

### 2.1. Cells

Madin-Darbey bovine kidney (MDBK) cells were provided by the National Bank for Industrial Microorganisms and Cell Cultures (Sofia, Bulgaria). Cells were cultivated in DMEM growth medium (Gibco, Grand Island, NY, USA) containing 10% fetal bovine serum (Gibco, Grand Island, NY, USA), 10 mM HEPES buffer (AppliChem GmbH, Darmstadt, Germany), and antibiotics 100 IU/mL penicillin and 100 μg/mL streptomycin (Sigma-Aldrich, Schnelldorf, Germany Cell incubation was carried out in a HERA cell 150 incubator (Heraeus, Hanau, Germany) at 37 °C with a 5% CO_2_ atmosphere, and they were propagated twice a week.

### 2.2. Viruses

Herpes simplex virus type 1, Victoria strain (HSV-1) was obtained from Prof. S. Dundarov at the National Center of Infectious and Parasitic Diseases (Sofia, Bulgaria). HSV-1 was replicated in confluent monolayers of MDBK cells using a maintenance solution of Dulbecco’s modified Eagles’s medium (DMEM) from Gibco BRL (Paisley, Scotland, UK) supplemented with 0.5% fetal bovine serum (Gibco BRL, Scotland, UK) and antibiotics (100 IU/mL penicillin and 100 μg/mL streptomycin). Following incubation at 37 °C in a 5% CO_2_ incubator, the viral yield was frozen at −80 °C. The infectious titer of the stock virus was determined to be 10^8.5^ CCID_50_/mL.

### 2.3. Light Source

The used light source was a light-emitting diode (LED) with an irradiation spectrum with a maximum of 660 nm (ELO Ltd., Sofia, Bulgaria), which overlapped the absorption spectra of the applied phthalocyanines (Figure 2). The samples were irradiated with a constant fluence rate of 50 mW/cm^2^ and light dose of 12 J/cm^2^. The fluence rate was constant as measured with a power meter PM 100D with a sensor S120VC (Thorlabs Inc., North Newton, KS, USA).

### 2.4. Irradiated and Cytotoxicity Assay

The cytotoxicity of the studied photosensitizers was determined without irradiation, (i) dark toxicity, and after irradiation (ii) phototoxicity. In the determination of dark toxicity, a monolayer cell culture of MDBK cells was inoculated in the dark with 0.1 mL of tenfold falling dilutions of the respective photosensitizer, and then, the cells were incubated in the dark in a humidified atmosphere at 37 °C and 5% CO_2_. At 48 h post-inoculation, cells were examined microscopically for visible signs of cytotoxicity, and the medium was removed, and the cells were washed and incubated with neutral red (Neutral Red Uptake Assay, [38]) at 37 °C for 3 h. After incubation, the neutral red dye was removed, and the cells were washed with PBS, and 0.15 mL/well desorbing solution (1% glacial acetic acid and 49% ethanol in distilled water) was added. The optical density (OD) of each well was read at 540 nm in a microplate reader (Biotek Organon, West Chester, PA, USA). The 50% cytotoxic concentration (CC_50Dark_) was defined as the concentration of applied compound that reduces the cell viability by 50% compared to the untreated controls. Each sample was tested in triplicate, with four wells per cell culture on a test sample. The maximum tolerable concentration (MTC_Dark_) was determined by the concentration at which the compound did not affect the cell monolayer. To evaluate the phototoxicity, the same experimental setup was applied, but the samples were irradiated immediately after inoculation for a period of 20 min at room temperature. The cytotoxic concentration of 50% under irradiation (CC_50LED660nm_) and the maximum tolerated concentration under irradiation (MTC_LED660nm_) were determined. The ratio between the two cytotoxic concentrations (in the dark and with irradiation) of each photosensitizer shows the photoinhibition factor (PIF). The PIF was determined according to Equation (1):PIF = CC_50Dark_/CC_50LED660nm_(1)

### 2.5. Determination of Infectious Viral Titers

The cells are cultivated in 96-well plates. After the formation of a confluent monolayer, the cells are infected with 0.1 mL viral suspension in tenfold falling dilutions by the endpoint dilution method [39]. After 1 h of adsorption, the non-adsorbed virus is removed, and to the cells is added 0.1 mL/well supporting medium. The plates are incubated at 37 °C and 5% CO_2_ in a HERA cell 150 CO_2_ incubator (Radobio Scientific Co., Ltd., Shanghai, China) for 2 days. Uninfected cells were used as the control, cultivated under the same conditions as the viral control (cells infected with the maximum concentration of the virus and demonstrate the maximum cytopathic effect). The infectious viral titer is determined by microscopic monitoring of the cellular monolayer and the determination of the cytopathic effect (CPE). Visually defined CPE is confirmed by coloring with neutral red, as described in Section 2.4. Viral titers are expressed as lg IU (infectious units) (CCID_50_)/0.1 mL.

### 2.6. Photodynamic Inactivation

#### 2.6.1. Effect on Extracellular Virions

Samples containing an equal (1:1) ratio of virus suspension (10^5^ CCID_50_) and the applied photosensitizer in its predetermined maximum tolerated concentration (MTC_LED660nm_) were prepared. The samples were irradiated for different periods of time (5, 10, 15, and 20 min) at room temperature. This was followed by determination of the viral titers in samples and controls to determine the residual viral infectivity in a sample. Three types of controls were tested: (i) virus control, i.e., not irradiated and not treated with a photosensitizer, (ii) a virus control that was irradiated but not treated with the photosensitizer, and (iii) a virus control that was not irradiated but treated with the photosensitizer. Photoinactivating activity is expressed as the difference between the infectious virus titer of the test sample and the controls (Δlgs), as is summarized in Table 1.

#### 2.6.2. Effect on Viral Adsorption

Two experimental setups were carried out to conduct this experiment.

##### First Experimental Setting

A monolayer cell culture of MDBK cells in 24-well plates was pre-cooled to 4 °C and inoculated simultaneously with 10^4^ CCID_50_ of HSV-1 and the tested photosensitizers in their MTK_LED660nm_ (presented in Table 1). Irradiation was performed on ice to maintain a low temperature of the cell monolayer. In this experimental setup, the samples were irradiated with a constant dose but for different time intervals of virus adsorption. Each time interval started from the time of virus inoculation to the cell monolayer and lasted 15, 30, 45, or 60 min. After the incubation, the different samples of cells were washed with PBS to remove the rest of photosensitizers and unattached virus. Then, they were covered with maintenance medium and incubated at 37 °C for 24 h. Along with the test sample containing irradiated photosensitizer and the virus, three types of virus controls were also prepared in parallel: (i) virus control, i.e., without light and photosensitizer, (ii) a light viral control that was only irradiated but not treated with the photosensitizer, and (iii) a dark viral control that was treated with the photosensitizer but not with light. After triple freezing and thawing of the obtained samples, the infectious virus titer of each sample (and control) was determined by the terminal dilution method. Differences in the viral titers compared to the viral control were reported, and ∆lg was calculated. Each sample was prepared in quadruplicate (Table 1).

##### Second Experimental Setting

The second experimental setup resembled the first but differed from it in the irradiation intervals. Again, the samples were irradiated for different time intervals of virus adsorption, but in the first one, the time started when the virus was inoculated to the cell in the presence of the photosensitizer and continued for 15 min of adsorption. In the second sample, the irradiation started from 16 to 30 min of adsorption, and the third sample from 31 to 45 min and, in the fourth sample, from 46 to 60 min of virus adsorption. Viral titers in the samples were determined, and ∆lgs was calculated when compared to the viral controls (Table 1).

#### 2.6.3. Effect on the Viral Replicative Cycle

##### Determination of the Antiviral Effect of the Tested Photosensitizers by Inhibition of the Cytopathic Effect and Staining with Neutral Red

A cytopathic effect inhibition (CPE) test was used to evaluate the antiviral activity of the tested photosensitizers. A confluent cell monolayer of MDBK cells in 96-well plates was infected with 100 cell culture infectious doses of 50% (CCID_50_) in 0.1 mL. After 60 min of virus adsorption, the non-attached virus was removed, and the photosensitizer was added at various concentrations. The samples with the photosensitizer were irradiated for a period of 20 min, and three types of controls were prepared in parallel, the same as described above. The cells were washed with PBS to remove the photosensitizer that was not uptaken. Cells were incubated for 48 h at 37 °C. The cytopathic effect was determined using a neutral red uptake assay, as described in Section 2.4., and the percentage of CPE inhibition for each concentration of the tested photosensitizer was calculated using the following Formula (2):% CPE = [OD_test sample_ − OD_virus control_]/[OD_toxicity control_ − OD_virus control_] × 100 (2)
where OD_test sample_ is the mean value of the ODs of the wells inoculated with the virus and treated with a photosensitizer in the respective concentration, OD_virus control_ is the mean value of the ODs of the virus control wells (without the photosensitizer in the medium), and OD_toxicity control_ is the mean value of the ODs of the wells not inoculated with virus but treated with the corresponding concentration of the test photosensitizers. The 50% photoinactivating concentration (PIC_50_) was defined as the concentration of the test photosensitizers that inhibited 50% of viral replication when compared to the virus control. The determination of PIC_50_ was carried out under conditions conducted in the dark (PIC_50Dark_) and under irradiation (PIC_50LED660nm_). The photoinactivating index (PII), which represents the selectivity of the investigated photosensitizers, was calculated from the ratio CC_50LED660nm_/PIC_50LED660nm_ (Table 1).

##### Determination of the Antiviral Effect of the Tested Photosensitizers by Inhibition of the Cytopathic Effect and Determination of the Infectious Virus Titers

A cell monolayer of MDBK cells in 24-well plates was inoculated with 10^3^ cell culture infectious doses of 50% (CCID_50_) in 1 mL of virus suspension. After 60 min of adsorption, the viral inoculum was removed, and the cells were washed with PBS. The maintenance medium was added and also different concentrations of the tested photosensitizer. The samples with the photosensitizer were irradiated for 20 min, then the same procedure described in Section Determination of the antiviral effect of the tested photosensitizers by inhibition of the cytopathic effect and staining with neutral red was followed. The photosensitizer was removed, and the cells were washed with PBS. The cells were incubated for 24 h at 37 °C. The samples were then frozen, and the infectious viral titer was determined by the final dilution method (Table 1).

### 2.7. Phthalocyanines and Other Chemicals

Two tetra-methylpyridiloxy substituted phthalocyanine complexes of zinc (ZnPcMe) and gallium (GaPcMe) were prepared as previously reported [40,41]. The chemical structures of both phthalocyanines are shown in Figure 1. Prior in vitro experiments with two stock solutions (2 mM) of the ZnPcMe and GaPcMe were prepared in dimethylsulphoxide (DMSO, Uvasol). The solutions were stored in closed vials covered with aluminum foil. The serial dilutions were carried out in sterile 0.05 M phosphate-buffered saline (PBS) so that the DMSO content was below 5% of the total volume. The exact concentration of compounds was monitored by the absorption spectra, which were recorded on a Shimadzu UV–Vis 3000, Osaka, Japan (Figure 2). All solvents and solids for synthesis and photo studies were purchased from Sigma-Aldrich, Schnelldorf, Germany and Merck & Co., Inc., Rahway, NJ, USA.

### 2.8. Statistical Analysis

Data on the cytotoxicity of the photosensitizers and antiviral effects were analyzed statistically. The values of CC_50_, IC_50_, PIC_50_, and Δlgs were presented as the mean ± SD. The significance of the differences between the cytotoxicity values of the photosensitizers and ACV was determined by Student’s *t*-test. The effect of each photosensitizer on extracellular virions and the stage of their adsorption was compared to the untreated viral control and presented statistically by Student’s *t*-test. When using Student’s *t*-test, *p*-values of <0.05 were considered significant. Data were analyzed by GraphPad Prism 8 software (San Diego, CA, USA).

## 3. Results

The study firstly explored the cytotoxicity of applied phthalocyanines on cell culture in order to exclude possible undesirable cytotoxicity of the studied photosensitizers. It was observed on the MDBK cell line in dark and light conditions. Cytotoxicity was determined in the dark, ensuring a nonactivated photosensitizer and, in the light, at LED 660 nm irradiation for 20 min. Both studied photosensitizers showed similar cytotoxicity, although ZnPcMe demonstrated slightly higher dark and phototoxicity. The concentration of a phthalocyanine compound that is cytotoxic to 50% death of the cell monolayer-both in the dark (CC_50Dark_) and under irradiation (CC_50LED660nm_) was determined (Table 2). The ratio between two values was calculated to determine the photoinhibition factor (PIF) of each photosensitizer. A higher value of PIF was shown for ZnPcMe (PIF = 15.9) and next for GaPcMe (PIF = 13.74), as presented in Table 2. The maximum tolerable concentration of the photosensitizers presented the values at which there was no visible effect on cell monolayer viability in the dark (MTC_Dark_) and light conditions (MTC_LED660nm_). The MTC values were determined for both photosensitizers with the same MTC value: MTC_Dark_ = 10.0 µg/mL and MTC_LED660nm_ = 1.0 µg/mL (Table 2). The antivirus experiments were performed with selected MTCs values.

The influence of both phthalocyanine photosensitizers on the infectivity of HSV-1 strain Victoria at different stages of its reproduction was studied. A general scheme of the antiviral experiments was carried out, and the periods of irradiation of the virus, depending on when the viral infection occurs, are presented in Table 1. First, the effect of PDI on the extracellular virions was examined before they had contact with the sensitive cells (MDBK). In Table 1, this is the irradiation period of Method 2.6.1. The virus and each of the photosensitizers were incubated for various time intervals in the dark and after irradiation in the absence of cells. The virus was incubated with ZnPcMe or GaPcMe for various time intervals in the dark and at irradiation in the absence of cells. The samples of the control groups were then titrated for evaluation of the viral infectivity and to determine the parameter Δlgs. The dark viral controls containing a photosensitizer but not irradiated and the light control with only light application but without a photosensitizer did not show a decrease in viral titer as compared to the control group without a photosensitizer and light. The data suggested the typical behavior of a photosensitizer, which was as followed: (1) the photosensitizer has no effect on the viral particle if it is not exposed to 660 nm light; (2) a direct irradiation of the virus does not lead to a decrease in its infectivity. ZnPcMe clearly inhibited the virus particles at the first time interval, which resulted in a decrease in the viral titer by Δlgs = 2.66. This effect was enhanced at a higher light dose and with the exposure time, respectively. Therefore, an increasing of the radiation dose can probably improve the effect. At the longest time interval (20 min), the effect was a decrease in the viral titer by Δlgs = 4.5. Considering the difference in the coordinated metal ion, GaPcMe at 5 min of exposure showed statistically negligible inhibition, but at 10 min of irradiation, the value increased significantly (Δlgs = 1.75). With an increase in the exposure time, the effect was also intensified. Although it was slightly weaker than that reported for ZnPcMe, it approached it-for 20 min of irradiation Δlgs = 4.25 (Table 3).

The observation that both studied photosensitizers showed a significant inhibitory effect on extracellular HSV-1 virions was a good base to check whether or not they can also influence the step of viral adsorption to susceptible MDBK cells. Again, the effect was evaluated for the different time intervals of irradiation within the viral adsorption time. In this experimental setup, the virus and photosensitizers are added simultaneously to the cell monolayer of MDBK cells. Two experimental setups were used, which were described in the Experimental Section 2.6.2 and presented in Table 1. In the first experimental setup, the exposure time was set to 15, 30, 45, and 60 min, starting from the beginning of the adsorption process. The irradiation time was of varying durations. The results showed that, in the first 15 min of viral adsorption, the process was significantly inhibited by both photosensitizers, which led to a decrease in viral titers by a value of Δlgs = 3.25. At an increase in the exposure time and the radiation dose, respectively, the effect was significantly enhanced, being much more noticeable for the samples treated with GaPcMe, with a high reduction in the viral titer at 60 min (Δlgs = 6.5) and lower reduction for ZnPcMe (Δlgs = 4.5), as is shown in Table 4.

The second experimental setup was also carried out with the simultaneous addition of the virus and the photosensitizer to the cells. The difference from the first experimental setup is that, here, the irradiations were for equal time intervals of 15 min but were distributed in a different time interval from the adsorption of the virus. The difference between the irradiation intervals in the two sets, Section First Experimental Setting and Section Second Experimental Setting, are presented in Table 1. The obtained results showed that both photosensitizers affected this stage of virus–cell interaction similarly, regardless of the moment of the viral adsorption and the start of irradiation. Both tested phthalocyanines showed a decrease in the viral titers at all test time intervals with a value Δlgs = 3.25 (Table 4).

The presented results showed that the studied photosensitizers have an effect on the extracellular virions, as well as an inhibitory effect on the stage of their adsorption to the cell. The evaluation of the PDI effect on HSV-1 replication at stage virus into the host cell suggested light dependence on the inhibition. Briefly, the experiment was carried out in the dark and under irradiation for 20 min after the end of viral adsorption, using the photosensitizers’ concentrations that were in the pre-established nontoxic range. During the irradiation period, extracellular virions and unadsorbed viruses were removed from the system. Irradiation was performed only on virus particles that had penetrated the sensitive cells. This study showed no inhibitory effect on the dark control group. The observed effect was done in two ways (Table 1; methods Determination of the antiviral effect of the tested photosensitizers by inhibition of the cytopathic effect and staining with neutral red and Determination of the antiviral effect of the tested photosensitizers by inhibition of the cytopathic effect and determination of the infectious virus titers). In the first method, the determination of the result was with the dye neutral red. A value of PIC_50_ of the studied photosensitizers was determined depending on their previously established CC_50LED660nm_, and the selectivity of their antiviral effect was determined by calculating their PII. ZnPcMe induced a remarkable photosensitivity, as seen for PII = 33.0, while, for GaPcMe, this effect was more modest with PII = 5.1 (Table 2). In the second method, the decrease in viral titer caused by photosensitizers in treated and irradiated samples was compared to the viral control. The effect of GaPcMe at concentrations of 1.0 μg/mL and 0.1 μg/mL was remarkable. A decrease in the viral titer with Δlgs = 4.0 was reported. However, ZnPcMe showed a weaker effect (Table 5)

## 4. Discussion

Photodynamic therapy (PDT) for application towards viruses is under an intensive development since the last pandemic time. PDT has been evaluated as an effective new approach to treat and prevent the spread of multiple viral infections. PDT by its nature is a gentle treatment procedure with rare side effects, which are the main concern of the currently in usage chemotherapy. Moreover, PDT leads to the minimal probability of the formation of resistant mutants, as a result of the long-term treatment with the same therapeutic drugs. Presently, the challenge in PDT is to find the right regime of administration of light–photosensitizer doses so that only the viral particles are affected but the host cells remain maximally unaffected, terminating the virus growth.

The clinical application of photodynamic inhibition (PDI) for the treatment of infectious diseases has so far mainly focused on the treatment of viral lesions [42,43]. The first application of PDI on viruses was reported in the early 1970s in the treatment of herpes infection [44]. The PDI procedures described so far have mainly been limited to the treatment of papillomatosis caused by human papillomavirus (HPV), such as laryngeal papillomatosis [45] and epidermodysplasia verruciformis [46] and, in rare cases, for the treatment of viral complications in AIDS patients [47,48]. Considerable progress has also been made in viral photodynamic disinfection of blood products [43,49]. Considering an effective photosensitizer, it must induce changes in specific viral components, such as the lipid envelope in the case of enveloped viruses, the protein envelope, or the viral nucleic acid [49,50]. The generation of singlet oxygen and other ROS during PDI leads to interactions with multiple enzymes, leading to inhibition of the protein synthesis [51] and molecular modification of DNA chains, which alters the transcription of genetic material during viral replication (mutagenic effect) [52,53,54,55].

Phthalocyanines are recognized as powerful photosensitizers through the Type II mechanism with high singlet oxygen generation [42]. The porphyrin-like compounds are known with the formation of singlet oxygen and small quantity of other ROS as the main species of photosensitization. The chemical structure is essential for anti-TBEV (tick-borne encephalitis virus) activity, as was shown in a recent study [56]. The presence of different structurally distinctive substituents and metal cations may significantly alter the antiviral activity of the main (porphyrin) macrocycle. The mechanism involves targeting the viral envelope and blocking the viral fusion machinery, which can efficiently inactivate multiple phylogenetically unrelated enveloped viruses [56]. The interaction with viral surface proteins cannot be excluded in the case of nonenveloped viruses. This study demonstrated extremely interesting mechanisms of antiviral activity of these compounds manifested at different stages of the viral life cycle. A recent study also confirmed the mechanism of the antiviral activity of the four porphyrins based on metal-organic framework to be the singlet oxygen generation induced by visible light exposure [57]. It was assumed that this highly reactive oxygen species (^1^O_2_) may further attack the phospholipids of the virus envelope, which prevent the virus entry into the cells.

A different structured compound such as a lipophilic thiazolidine derivative (LJ001) was also studied with the antiviral activity associated with light-dependent Type II photosensitization [58]. The study evaluated as the main target of singlet oxygen generation unsaturated phospholipids, as was proven by the formed hydroxylated fatty acids. The oxidation was determined by the allylic hydroxylation of unsaturated phospholipids and trans-isomerization of the double bond and parallel formation of a hydroxyl group in the lipid bilayer. The lipid oxidation influenced the viral membranes and the effectiveness of virus–cell membrane fusion. The authors also evidenced that LJ001 towards enveloped viruses targeted the membrane, causing the phospholipid modifications, lipid oxidation, and distress in the fine-tuned biophysical characteristics of viral membranes by increasing the membrane curvature and/or decreasing the fluidity. The considered mechanism of action explained the noncytotoxic effect to cells at antiviral LJ001 concentrations unless the ability of the cell to repair its membranes is compromised [59]. The results showed that singlet oxygen induced by LJ001 and analog compound JL103 led to some variation on the membrane properties, such as a reduced fluidity and odor, which resulted in a lowering of the membrane fusion [60]. A similar compound (LJ002) was proved without toxicity, both in vitro and in vivo [61]. The singlet oxygen produced by LJ002 was observed to performed oxidative changes in lipids in the envelope that destroyed the viral membrane, inhibiting viral and cell membrane fusion, which is needed for infection. A recent study with derivatives of perylenylethynyl-uracils showed pronounced activity and selectivity against all tested five enveloped viruses [62]. As was shown before, the antiviral activity of the compound was evaluated with efficiency for a singlet oxygen generation light dose, the membrane accumulation for viral inhibition. Similar results were obtained against SARS-CoV-2 tested with perylene derivatives, suggesting antiviral potency at red light application [63]. It was concluded that the photosensitization is the major mechanism of perylene-based compounds as antivirals towards multiple enveloped viruses, resulting in membrane rheology of virus.

Most known experiments demonstrate that enveloped viruses are significantly more sensitive to PDI than non-enveloped viruses [64,65] because of the structural disruptions of the envelope and loss of virus infectivity [66,67,68,69,70]. This indicates that the main PDI targets of action are the unsaturated lipids and structural proteins included in the envelope composition [53,71,72,73,74]. The studies showed that the main damage in proteins is the formation of protein cross-links, followed by other types of damage that include changes in the protein’s molecular conformation, mass, and charge [43]. The mechanism of cell death includes the photooxidation of sensitive amino acid residues such as cysteine, L-histidine, tyrosine, methionine, and tryptophan and the next covalent cross-linking of the peptide chains [75]. The last is leading to the formation of molecular aggregates [43,76,77] and disruption of their normal folding conformation, which affects their normal functioning [78].

The studies suggested that HSV-1 viral envelope proteins are the major targets of the photosensitization with merocyanine 540 [79]. Some phthalocyanine derivatives have also been shown to induce cross-links in the HSV protein, resulting in a loss of infectivity [43,80]. Protein analysis by SDS-PAGE after PDI treatment with phthalocyanine derivatives revealed irreversible changes in the envelope proteins of HSV-1. The appearance of cross-linked material on top of the gel and the changes in the molecular mass and molecular charge of the proteins were observed. It is accepted that these changes most likely contributed to the inactivation of HSV-1 [73].

Both studied phthalocyanine complexes of Zn(II) and hydroxy-Ga(III) differ in the coordinated metal ion into ligand molecules (Figure 1). The photophysical properties, such as a strong Q-band in the far-red region (~680 nm), showed a typical spectrum of non-aggregated phthalocyanines in water medium only for GaPcMe (Figure 2). An appropriate value for a photosensitizer with a fluorescence quantum yield of 0.33 (ZnPcMe) and 0.25 (GaPcMe) was reported [81]. The efficacy of singlet oxygen generation, as was previously evaluated, showed a sufficient quantum yield of 0.41 (ZnPcMe) and 0.36 (GaPcMe). The photodynamic results on herpes virus assumed the mechanism of singlet oxygen (Type II) in the control for the antiviral action of the applied phthalocyanines. A small contribution of hydroxyl radical and superoxide ion (Type I reaction) may also be involved in the oxidation of this lipid-enveloped virus. The difference in the antiviral inhibition with ZnPcMe and GaPcMe may be explained with the difference in the photochemical properties and photosensitization capability of the compounds. The big atom of gallium leads to some steric hindrance and better PDT efficacy on the target cells [82]. Zn(II) phthalocyanine demonstrated a significant virucidal activity against Herpes virus, even in low concentrations and at irradiation of 5 min. Smetana et al. [69], in an early study on anti-herpes virus activity of different phthalocyanines, showed an extremely high efficiency of different derivatives. This study suggested the unusual damage of the viral envelope, which prevented further viral adsorption and its penetration in the host cells. A similar result was obtained by application of hypericin as photosensitizer [83]. 

The obtained results in the present study with ZnPcMe and GaPcMe against the infectivity of HSV-1 strain Victoria are in agreement with previous studies concerning PDI with phthalocyanines. Our study reported remarkable activity against extracellular HSV-1 virions after PDI, as was described about the virucidal inactivation in the previous work [84]. The suggestion for the strong virucidal effect, which we report with a decrease in the viral titer more than Δlgs > 4, is because of the damage of the lipid envelope and the specific viral proteins. Another supposition for the even stronger inhibitory effect, which we observed, may be related to the virus’ adsorption step. Most likely, the viral proteins needed to recognize the sensitive cells are modified and cannot perform their function. It is also possible to partially modify the cellular receptors that recognize viral particles and allow the virus to enter the cell. This may be the crucial step that was preventing the virus from binding to the cell and penetrating it. Similar results were presented by other authors who studied PDI of herpes viruses with phthalocyanine derivatives [69].

We hypothesize that modification of the specific viral proteins that enter the cell after virus penetration and uncoating is the suggestion for the effect we report at the 20-min activation of the photosensitizers after completion of virus adsorption and cell penetration. This is too early a stage for viral expression to have actually started, but it is a time when specific viral proteins can be influenced and modified to block the expression of the earliest genes, thereby blocking further stages of viral replication.

The potential in antiviral PDT is considered with good perspectives because of the strong interdependence between light and the immune system [84]. The most used medical application of the method against viruses is now the photo-decontamination field of blood products. However, with the application of aminolaevulinic acid (ALA) against HPV indicators, the inactivation of viruses has also found entrance into clinical practice [49].

## 5. Conclusions

The study presents a remarkable inhibition of the extracellular virions of HSV-1 strain Victoria because of PDI with two phthalocyanines (ZnPcMe and GaPcMe). This was achieved in the case of an initial attachment step of the virions to the susceptible MDBK cells by application of ZnPcMe and GaPcMe and irradiation with a specific spectrum of irradiation (LED 660 nm). The second arrangement of the experiment with PDI application after the virus entered the cell resulted in a remarkable effect, especially for PDI with ZnPcMe. These observation leads to a hypothesis that the observed effects are due to damage in the lipid membrane and structural proteins of the virus, as well as the specific viral enzymes that are involved to initiate viral replication.

## Figures and Tables

**Figure 1 viruses-16-01937-f001:**
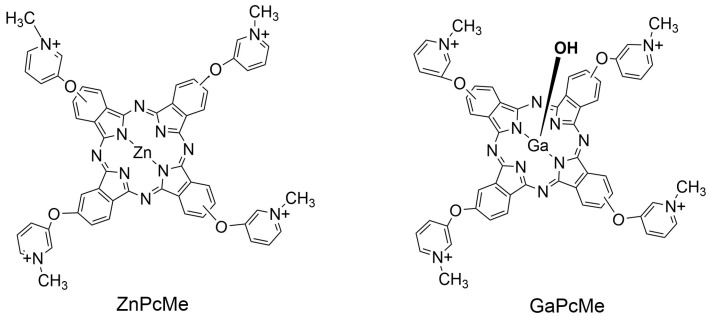
Phthalocyanine complexes (ZnPcMe and GaPcMe) used in the antiviral study on HSV-1.

**Figure 2 viruses-16-01937-f002:**
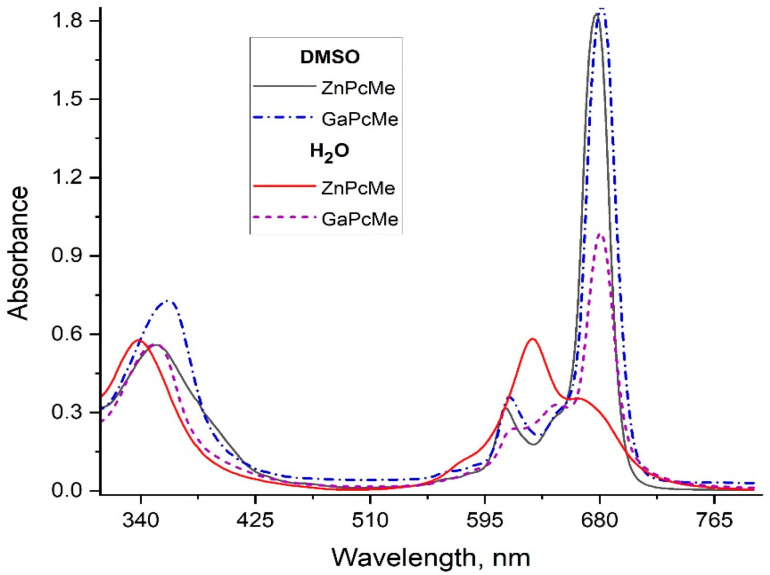
Absorption spectra of phthalocyanines in dimethylsulphoxide and water (10^−5^ M).

**Table 1 viruses-16-01937-t001:** Photodynamic inactivation.

Method	Stage of Influence of Viral Particles
Extracellular Virions		Virus Adsorption to Cells		Viral Replicative Cycle
Section 2.6.1	Room temperature	5 min	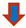	37 °C for 60 min	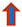	37 °C for 24 h
		10 min Room temperature	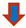	37 °C for 60 min	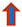	37 °C for 24 h
	15 min Room temperature	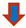	37 °C for 60 min	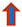	37 °C for 24 h
20 min Room temperature	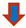	37 °C for 60 min	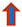	37 °C for 24 h
Section 2.6.2		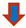	15 min4 °C		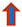	37 °C for 24 h
	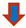	30 min 4 °C			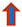	37 °C for 24 h
	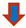	45 min 4 °C		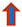	37 °C for 24 h
	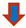	60 min 4 °C	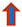	37 °C for 24 h
Section 2.6.2		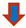	15 min 4 °C		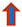	37 °C for 24 h
	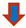		15 min4 °C		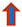	37 °C for 24 h
	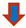		15 min4 °C		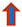	37 °C for 24 h
	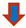		15 min4 °C	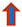	37 °C for 24 h
Section 2.6.3		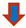		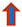	20 min 37 °C	37 °C for 24 h

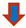
—addition of the viral inoculum. 
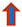
—removal of unadsorbed virus. 

—period of irradiation.

**Table 2 viruses-16-01937-t002:** Cytotoxicity on the MDBK cell line and effect on HSV-1 (Victoria strain) replication of both photosensitizers.

Photosensitizer	Cytotoxicity					Effect on Viral Replication		
	Dark		Light (LED 660 nm)			Dark	Light (LED 660 nm)	
	CC_50Dark_ Mean ± SD (μg/mL)	MTC_Dark_ (μg/mL)	CC_50LED660nm_ Mean ± SD (μg/mL)	MTC_LED660nm_ (μg/mL)	PIF	IC_50Dark_ (μg/mL)	PIC_50LED660nm_ Mean ± SD (μg/mL)	PII
ZnPcMe	52.4 ± 3.4 ***	10.0 ***	3.3 ± 0.6	1.0	15.9	-	0.1 ± 0.009	33.0
GaPcMe	63.2 ± 4.2 ***	10.0 ***	4.6 ± 0.8	1.0	13.74	-	0.9 ± 0.01	5.1
ACV	873.4 ± 6.8	320.0	-	-	-	1.2 ± 6.8	-	-

SD—standard deviation; *** *p* ˂ 0.001, comparing the value for each photosensitizer with ACV; CC_50_—50% cytotoxic concentration; MTC—maximum tolerable concentration; IC_50_—50% inhibitory concentration; PIC_50_—50% photoinactivating concentration; light-emitting diode (LED); PIF—photoinhibition factor; PII—photoinactivating index.

**Table 3 viruses-16-01937-t003:** Effect of photosensitizers on extracellular HSV-1 virions (Victoria strain).

Photosensitizer	Δlgs			
	5 min	10 min	15 min	20 min
ZnPcMe	2.66 ± 0.25 *	2.66 ± 0.25 *	3.5 ± 0.12 **	4.5 ± 0.25 **
GaPcMe	0.75 ± 0.12	1.75 ± 0.22 *	3.0 ± 0.12 **	4.25 ± 0.33 **

* *p* ˂ 0.05, comparing the effect of each photosensitizer with the viral control; Student’s *t*-test; ** *p* ˂ 0.01, comparing the effect of each photosensitizer with the viral control; Student’s *t*-test.

**Table 4 viruses-16-01937-t004:** Effect of photosensitizers on the viral adsorption step of HSV-1 (Victoria strain) to susceptible MDBK cells.

Compounds	Δlgs			
	First Experimental Setting			
	0–15 min	0–30 min	0–45 min	0–60 min
ZnPcMe	3.25 ± 0.10 **	3.25 ± 0.25 **	4.5 ± 0.30 **	4.5 ± 0.22 **
GaPcMe	3.25 ± 0.20 **	4.0 ± 0.12 **	6.0 ± 0.33 **	6.5 ± 0.35 **
	**Second Experimental Setting**			
	**0–15 min**	**16–30 min**	**31–45 min**	**46–60 min**
ZnPcMe	3.25 ± 0.12 **	3.25 ± 0.22 **	3.25 ± 0.15 **	3.25 ± 0.12 **
GaPcMe	3.25 ± 0.15 **	3.25 ± 0.20 **	3.25 ± 0.15 **	3.25 ± 0.12 **

** *p* ˂ 0.01, comparing the effect of each photosensitizer with the viral control; Student’s *t*-test.

**Table 5 viruses-16-01937-t005:** Antiviral effect of the tested photosensitizers by inhibition of the cytopathic effect and determination of the infectious virus titers.

Photosensitizer Concentration (μg/mL)	Δlgs	
	ZnPcMe	GaPcMe
1.0	1.5 ± 0.1 *	4.0 ± 0.2 **
0.1	1.25 ± 0.09 *	4.0 ± 0.12 **
0.01	0.5 ± 0.04	1.5 ± 0.08 *
0.001	0.5 ± 0.07	0.5 ± 0.06

* *p* ˂ 0.05, comparing the effect of each photosensitizer with the viral control; Student’s *t*-test; ** *p* ˂ 0.01, comparing the effect of each photosensitizer with the viral control; Student’s *t*-test.

## Data Availability

The raw data supporting the conclusions of this article will be made available by the authors on request.

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
