# Peer review of "Photodynamic Inactivation of Human Herpes Virus In Vitro with Ga(III) and Zn(II) Phthalocyanines"

_viruses, 2024, doi:10.3390/v16121937_

Round 1
Reviewer 1 Report (Previous Reviewer 2)
Comments and Suggestions for Authors
I would like to commend the authors for addressing all of my concerns. Well done on an interesting story.
Author Response
Comments 1: I would like to commend the authors for addressing all of my concerns. Well done on an interesting story.
Response 1: We thank the esteemed reviewer for his comments and suggestions for improving our manuscript. We thank him for the positive assessment he gave to our research.
Reviewer 2 Report (Previous Reviewer 1)
Comments and Suggestions for Authors
The revised version of the manuscript is significantly improved. However, I suggest a revision in the data presentation. The experimental settings in Table 1 are difficult to link to the obtained results. To improve clarity, I recommend adding a brief explanation of the methods used within the Results section.
Author Response
Comments 1: The revised version of the manuscript is significantly improved. However, I suggest a revision in the data presentation. The experimental settings in Table 1 are difficult to link to the obtained results. To improve clarity, I recommend adding a brief explanation of the methods used within the Results section.
Response 1: We thank the esteemed reviewer for his comments and recommendations that improved our manuscript. We have included a brief methodological explanation before presenting the results obtained by each different methodology.
This manuscript is a resubmission of an earlier submission. The following is a list of the peer review reports and author responses from that submission.
Round 1
Reviewer 1 Report
Comments and Suggestions for Authors
The manuscript by Vilhelmova-Ilieva et al. reports in vitro photodynamic inactivation of herpes simplex virus type1 by two known compounds, zinc and gallium phthalocyanines decorated with cationic pyridilium groups.
The title: “antiviral … inactivation of … virus”, a lame phrase, “… quaternized Ga(III) and Zn(II) phthalocyanines”, a misleading phrase, because quaternized are side chain pyridine residues, not a phthalocyanine core.
List of authors: “3 and 4” – bad presentation.
Page 2: “The structural features of antiviral PSs have to be directed to the inhibition of the viral reproduction in host cells” – a blurred statement without examples of particular structural features and their involvement in mechanisms of inhibition.
The structures of compounds (Figure 1): positions of pyridiloxy substituents is excessively specified, one nitrogen atom in the phthalocyanine core is missing, the hydroxyl on Ga atom is badly located.
Reference list: non-uniform font size, mixed style in journal abbreviations.
For better comprehension, the experiments from section 2.6 should be illustrated with diagrams showing time/temperature/inoculation/irradiation. Generally the description and discussion of time-of-addition experiments are scattered and hard to comprehend, so the experiments should be summarized in general scheme.
Although some discussion on possible mechanism of photosensitized inhibition of viral reproduction is present, it is rather general, e.g. citing the 1992 article (ref. 69) on photodamaging of HSV-1 envelope proteins. However, more recent studies suggest that unsaturated membrane lipids of a viral envelope are effective targets for singlet oxygen produced by photosensitizers (10.1371/journal.ppat.1003297, 10.1042/BJ20131058, 10.1016/j.redox.2020.101601, 10.1016/j.antiviral.2022.105508, 10.1016/j.virusres.2023.199158, 10.1016/j.antiviral.2023.105767, 10.1016/j.mtbio.2024.101165). Therefore, the compounds studied should discussed in terms of their ability to produce singlet oxygen.
Author Response
Comments 1: The title: “antiviral … inactivation of … virus”, a lame phrase, “… quaternized Ga(III) and Zn(II) phthalocyanines”, a misleading phrase, because quaternized are side chain pyridine residues, not a phthalocyanine core.
Response 1: The title has been edited.
Comments 2: List of authors: “3 and 4” – bad presentation.
Response 2: A correction has been made.
Comments 3: Page 2: “The structural features of antiviral PSs have to be directed to the inhibition of the viral reproduction in host cells” – a blurred statement without examples of particular structural features and their involvement in mechanisms of inhibition.
Response 3: The effect of photosensitizers on viral infection inside cells is poorly studied and there is no real evidence in the literature on which structure they exert an inhibitory effect – it is all supposition. Most experiments have been conducted with virions or on their adsorption stage to the host cell. Indeed, our statement is not well formulated based on the known information, so it has been removed.
Comments 4: The structures of compounds (Figure 1): positions of pyridiloxy substituents is excessively specified, one nitrogen atom in the phthalocyanine core is missing, the hydroxyl on Ga atom is badly located.
Response 4: Corresponding corrections have been made to the figure.
Comments 5: Reference list: non-uniform font size, mixed style in journal abbreviations.
Response 5: In the reference list, the font size and style of journal abbreviations have been standardized.
Comments 6: For better comprehension, the experiments from section 2.6 should be illustrated with diagrams showing time/temperature/inoculation/irradiation. Generally the description and discussion of time-of-addition experiments are scattered and hard to comprehend, so the experiments should be summarized in general scheme.
Response 6: For a better understanding of the experiments of Section 2.6, Table 1 was compiled, showing the time and temperature of the experiment, the inoculation and removal of the virus, and the period of irradiation.
Comments 7: Although some discussion on possible mechanism of photosensitized inhibition of viral reproduction is present, it is rather general, e.g. citing the 1992 article (ref. 69) on photodamaging of HSV-1 envelope proteins. However, more recent studies suggest that unsaturated membrane lipids of a viral envelope are effective targets for singlet oxygen produced by photosensitizers (10.1371/journal.ppat.1003297, 10.1042/BJ20131058, 10.1016/j.redox.2020.101601,10.1016/j.antiviral.2022.105508,10.1016/j.virusres.2023.199158, 10.1016/j.antiviral.2023.105767, 10.1016/j.mtbio.2024.101165). Therefore, the compounds studied should discussed in terms of their ability to produce singlet oxygen.
Response 7: We appreciate the suggestion to expand the discussion on the mechanism of action, especially regarding the ability of photosensitizers to produce singlet oxygen. Additional information was included in the discussion and the issue was addressed in more detail.
Reviewer 2 Report
Comments and Suggestions for Authors
Herpes simplex viruses are ubiquitous pathogens for which latency-reactivation cycles are a continued source of viral shedding and disease burden. Nucleoside analogues acyclovir and associated derivatives are therapeutically effective, however novel strategies are needed to circumvent viral resistance. To this end Vilhelmova-Ilieva et al. evaluated the effectiveness of photodynamic inactivation of HSV-1 adsorption to a Bovine cell line with ZnPcMe and GaPcMe.
Specific comments:
-Lines 34-35: This passage “…causes a range of clinical manifestations, including oral and genital herpes, keratitis, encephalitis, and disseminated neonatal disease” is copied directly from the abstract of reference #1 (Bautista, 2024). Authors should rewrite this passage and consider evaluating the entirety of the manuscript to ensure this was the only instance.
-Lines 73-74: May want to elaborate on what is meant by “disrupt the functionality of viral pathogen”. Specifically, what is meant by “functionality”? Is the viral envelope, capsid, or genomic material altered?
-Lines 97-104: Why were MDBK (bovine) used in experiments (other than generating viral stock) and not a relevant human cell line that is typically used in HSV-1 research? Such as MRC-5, HFF, LUHMES, etc.
-Lines 123-148: Would suggest including data from the neutral red cytotoxicity assay into the manuscript for the purpose of transparency. Important to ensure the restriction of HSV-1 is not due to a decrease in cell viability from ZnPcMe or GaPcMe treatment.
-Lines 164-173, 185-189, 208-211: When calculating the “photoinactivating activity” (delta Ig) which of the three controls (no irradiation + no photosensitizer; irradiated + no photosensitizer; or no irradiation + photosensitizer) is used in the comparison? It’s also important to note whether DMSO (carrier for photosensitizer) was used in the “not treated with the photosensitizer” group, if not these experiments should be repeated.
-Lines 202-229: There are two questions for this passage:
#1) When evaluating the viral replicative cycle, CCID50 (viral titers) should have been determined rather than a neutral red cytoxicity assay. This is a concern given the potential cellular toxicity of the photosensitizer and irradiation treatment. Would suggests repeating this experiment by directly measuring viral replicative cycle via CCID50 (viral titers).
#2) Was the viral adsorption, photosensitizer treatment and irradiation completed on ice before shifting to 37 C? If all these steps were completed on ice, then the assay is not measuring “effect on the replicative cycle” but viral entry into the cell. At temperatures of 4 C, HSV-1 will attach to the cell but not enter (Farnham and Newton, Virology 1959, PMID:13669316) and therefore the photosensitizer has the chance to inhibit the virus particle prior to entry, which appears to not be the intended purpose of this experiment. A more thorough experiment would be after the 1-hour viral adsorption (37 C to ensure attachment and entry is complete) is to then treat cells with the photosensitizer + irradiate and evaluate viral transcript/protein expression and viral DNA replication during the course of the lytic replication cycle (2, 6, 8, 12 hpi, etc). An experiment in this format would enable the determination at what stage of the HSV-1 replication cycle (other than adsorption) is impacted (Immediate Early gene expression, Early gene expression, viral DNA replication, Late gene expression).
-Tables 2, 3: Standard deviations are not included.
-Tables 1, 2, 3: No statistics are provided in this manuscript even though it was concluded that “GaPcMe at 5 min of exposure provoked the inhibition which was statistically negligible, but becomes significant at 10 min irradiation (Lines 284-285)”. Important to quantify the level of significance in the data.
-Lines 364, 366, 374: Change “explanation” to “suggests” given that this manuscript provides no primary mechanistical data for the activity of ZnPcMe and GaPcMe in suppressing HSV-1 adsorption.
Editorial comments:
-Line 4: Is an author name missing?
-Lines 34-36: Provide the primary reference(s) that reflects the clinical manifestations of HSV-1 rather than a basic (nonclinical) research article.
-Line 40: Change “supercapsid” to “envelope”.
-Lines 40-41: HSV-1 established latency within sensory neurons which includes both the dorsal root ganglia and the trigeminal ganglia.
-Lines 40-42: Consider rewriting. Latency is not a period by which HSV-1 is in a “non-infectious state”, but a period by which no lytic viral transcripts/proteins are expressed except for the Latency-associated transcript (LAT).
-Lines 248-250: Consider rewriting in a complete sentence. Difficult to follow the purpose of the experiment.
-Check for spelling (“adsorby” -line 152; etc) and word choice (“uninfected” not “unintended” -line 155; etc).
-Lines 322-323: How is “chemotherapy” relevant for an “application towards viruses”?
-Line 377: Change “enzymes” to “proteins”.
-Line 377: Change “replication” to “expression”.
Comments on the Quality of English LanguageFor this draft of the manuscript there are areas in which the proficiency of written English must be addressed. Meaning the text at times is difficult to follow in regard to how some of the experiments were performed, description of results, and the overall discussion/conclusions. Would suggest an editorial service to aid in improving the clarity of the English text. See the “editorial comments” for suggestion for the correct use of virology terms. These are only examples.
Author Response
Specific comments:
Comments 1: Lines 34-35: This passage “…causes a range of clinical manifestations, including oral and genital herpes, keratitis, encephalitis, and disseminated neonatal disease” is copied directly from the abstract of reference #1 (Bautista, 2024). Authors should rewrite this passage and consider evaluating the entirety of the manuscript to ensure this was the only instance.
Response 1: The passage was rewritten and the manuscript reviewed for other similar repetitions.
Comments 2: Lines 73-74: May want to elaborate on what is meant by “disrupt the functionality of viral pathogen”. Specifically, what is meant by “functionality”? Is the viral envelope, capsid, or genomic material altered?
Response 2: The passage was rewritten.
Comments 3: Lines 97-104: Why were MDBK (bovine) used in experiments (other than generating viral stock) and not a relevant human cell line that is typically used in HSV-1 research? Such as MRC-5, HFF, LUHMES, etc.
Response 3: We've used the MDBK cell line for HSV-1 replication because it's proven to be one of the best model systems in our team for probably over 20 years. I present cited and other authors who have preferred MDBK cells for the cultivation of herpes viruses:
А) Barreca C, O'Hare P. Suppression of herpes simplex virus 1 in MDBK cells via the interferon pathway. J Virol. 2004 Aug;78(16):8641-53. doi: 10.1128/JVI.78.16.8641-8653.2004. PMID: 15280473; PMCID: PMC479083.
В) Bartoletti AM, Tognon M, Manservigi R, Mannini-Palenzona A. Characterization of virus obtained from MDBK cells persistently infected with a variant of herpes simplex virus type 1 strain MP [HSV-1(MP)]. Virology. 1985 Mar;141(2):306-10. doi: 10.1016/0042-6822(85)90263-6. PMID: 3002019.
С) P.S. NARUTE1 , A.A. RAUT1 , M. SAINI2 , A. RAI1 , P.K. GUPTA. Inhibition of Bovine herpesvirus multiplication in MDBK cells by small interfering RNAs. Acta virologica. 2009, 53: 203 – 206. doi:10:4149/av_2009_03_203
Comments 4: Lines 123-148: Would suggest including data from the neutral red cytotoxicity assay into the manuscript for the purpose of transparency. Important to ensure the restriction of HSV-1 is not due to a decrease in cell viability from ZnPcMe or GaPcMe treatment.
Response 4: Data from the neutral red dye assay are presented in Table 2. It first presents the cytotoxicity of the tested substances - measured in the dark and under light conditions. Antiviral experiments were subsequently carried out at non-toxic concentrations of the substances, so that their cytotoxicity was excluded. This is precisely the purpose of determining their cytotoxicity.
Comments 5: Lines 164-173, 185-189, 208-211: When calculating the “photoinactivating activity” (delta Ig) which of the three controls (no irradiation + no photosensitizer; irradiated + no photosensitizer; or no irradiation + photosensitizer) is used in the comparison? It’s also important to note whether DMSO (carrier for photosensitizer) was used in the “not treated with the photosensitizer” group, if not these experiments should be repeated.
Response 5: In the three types of experiments, Δlg was calculated relative to the control in which the highest viral titer should be obtained, that is, the one that was not irradiated and was not treated with photosensitizers (the cytopathic effect that the virus would show without any external intervention). On the other hand, this control was compared to the other two types of controls (results not shown because Δlg = 0), but these were needed to demonstrate that 1) the photosensitizer did not affect the virus except during irradiation; 2) the virus is not affected by the irradiation itself and the results obtained are from the action of the photosensitizer.
DMSO is a standard solvent for many sparingly soluble substances. Although its cytotoxicity is well known to us and the tested dilutions of the studied photosensitizers are in the non-toxic range for MDBK cells of DMSO, it was also included as a control in the study to rule out any influence of it.
Comments 6: Lines 202-229: There are two questions for this passage:
#1) When evaluating the viral replicative cycle, CCID50 (viral titers) should have been determined rather than a neutral red cytoxicity assay. This is a concern given the potential cellular toxicity of the photosensitizer and irradiation treatment. Would suggests repeating this experiment by directly measuring viral replicative cycle via CCID50 (viral titers).
#2) Was the viral adsorption, photosensitizer treatment and irradiation completed on ice before shifting to 37 C? If all these steps were completed on ice, then the assay is not measuring “effect on the replicative cycle” but viral entry into the cell. At temperatures of 4 C, HSV-1 will attach to the cell but not enter (Farnham and Newton, Virology 1959, PMID:13669316) and therefore the photosensitizer has the chance to inhibit the virus particle prior to entry, which appears to not be the intended purpose of this experiment. A more thorough experiment would be after the 1-hour viral adsorption (37 C to ensure attachment and entry is complete) is to then treat cells with the photosensitizer + irradiate and evaluate viral transcript/protein expression and viral DNA replication during the course of the lytic replication cycle (2, 6, 8, 12 hpi, etc). An experiment in this format would enable the determination at what stage of the HSV-1 replication cycle (other than adsorption) is impacted (Immediate Early gene expression, Early gene expression, viral DNA replication, Late gene expression).
Response 6:
#1) The neutral red assay allows the determination of the photoinhibitory index of each photosensitizer. This index excludes the cytotoxicity of the given substance, because it is calculated from CC50 LED 660 nm / PIC50 LED 660 nm and represents the selective action of the antiviral substance towards the replicative cycle of the specific virus. We cannot give up the results of this experiment. However, we accept that many researchers prefer to determine the antiviral effect by monitoring the decrease in viral titers. Therefore, we accepted your remark and performed the experiment again, but this time it was reported by decreasing viral titers. The new results are included in the new version of the manuscript as section 2.6.3.2. The obtained results are presented in Table 5.
#2) Irradiation of the photosensitizer during virus adsorption was carried out on ice, and the cell plates were previously cooled in a refrigerator. This experiment does not track the effect on the replicative cycle, only the effect on the adsorption (not penetration) step of the virus to the cell. That is, the goal is to track whether the photosensitizer blocks the attachment of the virus to the cell when it is still outside the cell. What you propose after one hour of adsorption at 37 ° C to be the treatment and irradiation is what we have done in methodology 2.6.3. and we have used neutral red staining (to rule out cytotoxicity of photosensitizers). The aim of the study was to determine whether there is an influence on the internal replicative cycle at all, because the data in the literature are scarce and most of the data are related to the influence of photosensitizers on the extracellular virions or their adsorption stage. We used only one irradiation time interval, which in our previous studies proved to be optimal. But really, our next research will follow which stage of the HSV-1 replication cycle the photosensitizers affect the most. The study will be extended to other virus families.
Comments 7: Tables 2, 3: Standard deviations are not included.
Response 7: We thank the esteemed reviewer for the pointed out omission. Standard deviations are included.
Comments 8: Tables 1, 2, 3: No statistics are provided in this manuscript even though it was concluded that “GaPcMe at 5 min of exposure provoked the inhibition which was statistically negligible, but becomes significant at 10 min irradiation (Lines 284-285)”. Important to quantify the level of significance in the data.
Response 8: Thank you for your comment. The statistics have been supplemented in the indicated tables.
Comments 9: Lines 364, 366, 374: Change “explanation” to “suggests” given that this manuscript provides no primary mechanistical data for the activity of ZnPcMe and GaPcMe in suppressing HSV-1 adsorption.
Response 9: The correction has been made.
Editorial comments:
Comments 1: Line 4: Is an author name missing?
Response 1: There is no missing author, just a misspelling. The correction has been made.
Comments 2: Lines 34-36: Provide the primary reference(s) that reflects the clinical manifestations of HSV-1 rather than a basic (nonclinical) research article.
Response 2: Primary references reflecting the clinical manifestations of HSV-1 are indicated.
Comments 3: Line 40: Change “supercapsid” to “envelope”.
Response 3: The correction has been made.
Comments 4: Lines 40-41: HSV-1 established latency within sensory neurons which includes both the dorsal root ganglia and the trigeminal ganglia.
Response 4: The correction has been made.
Comments 5: Lines 40-42: Consider rewriting. Latency is not a period by which HSV-1 is in a “non-infectious state”, but a period by which no lytic viral transcripts/proteins are expressed except for the Latency-associated transcript (LAT).
Response 5: The sentence has been rewritten.
Comments 6: Lines 248-250: Consider rewriting in a complete sentence. Difficult to follow the purpose of the experiment.
Response 6: The sentence has been rewritten.
Comments 7: Check for spelling (“adsorby” -line 152; etc) and word choice (“uninfected” not “unintended” -line 155; etc).
Response 7: The suggested corrections have been made.
Comments 8: Lines 322-323: How is “chemotherapy” relevant for an “application towards viruses”?
Response 8: As we have mentioned in the text, the clinical application of photodynamic inhibition has focused mainly on the treatment of superficial viral lesions. The first such application was reported in the early 1970s in the treatment of herpes infection. The procedures described so far are mainly limited to the treatment of papillomatosis caused by human papillomavirus, such as laryngeal papillomatosis and epidermodysplasia verruciformis, and in some cases, for the treatment of viral complications in AIDS patients. With appropriate development of PDI, it is possible to actually introduce it into the practice of treating various skin infections. Given the good results that PDI gives in herpes simplex virus, it could also be introduced in more severe cases of varicella zoster virus infections, and in severe relapses, it could lead to a shortening of the duration of symptoms by several weeks.
Comments 9: Line 377: Change “enzymes” to “proteins”.
Response 9: The suggested corrections have been made.
Comments 10: Line 377: Change “replication” to “expression”.
Response 10: The suggested corrections have been made.
Comments 11: Comments on the Quality of English Language
For this draft of the manuscript there are areas in which the proficiency of written English must be addressed. Meaning the text at times is difficult to follow in regard to how some of the experiments were performed, description of results, and the overall discussion/conclusions. Would suggest an editorial service to aid in improving the clarity of the English text. See the “editorial comments” for suggestion for the correct use of virology terms. These are only examples.
Response 10: The English language was revised throughout the manuscript.
Reviewer 3 Report
Comments and Suggestions for Authors
The manuscript presents a study of two tetra-substituted gallium and zinc phthalocyanines (ZnPcMe and GaPcMe) applied as photosensitizers in photodynamic inactivation of human herpes simplex virus type 1 (HSV-1) at different stages of its reproduction. The topic is both interesting and relevant, but certain questions have remained open and there are minor errors that need to be corrected.
In this study, two different metallophthalocyanines were used and the only difference in those two structures is thus two different metal ions used for chelation (gallium and zinc). The discussion lacks an analysis of the PDI mechanism for these two compounds and the connection of their (photo)physical characteristics with the obtained results of their biological activity. It would be particularly useful to link the production of reactive oxygen species/singlet oxygen with the obtained biological results and specific differences between two photosensitizers.
Based on the obtained results it would be useful to discuss how this PDI could be applied in practice.
Check the structures in Figure 1 – one nitrogen is missing from the macrocycle (on the right in both structures). Please check bonds with hydroxy group in gallium phthalocyanine.
There are no units for concentration in Table 1.
Grammar should be checked and corrected throughout the manuscript (e.g. line 140 ‘…were propagate…’, line 330: ‘PDI procedure described so far has mainly be limited to the treatment’ etc.).
Author Response
Comments 1: In this study, two different metallophthalocyanines were used and the only difference in those two structures is thus two different metal ions used for chelation (gallium and zinc). The discussion lacks an analysis of the PDI mechanism for these two compounds and the connection of their (photo)physical characteristics with the obtained results of their biological activity. It would be particularly useful to link the production of reactive oxygen species/singlet oxygen with the obtained biological results and specific differences between two photosensitizers.
Response 1: We are grateful for the suggestion to discuss the obtained biological activities in relation to the structure of the two photosensitizers, as well as to pay more attention to the formation of reactive oxygen species/singlet oxygen as a result of the photodynamic reaction. Additional information is included in the discussion of the manuscript and the issue is considered in more depth.
Comments 2: Based on the obtained results it would be useful to discuss how this PDI could be applied in practice.
Response 2: The recent pandemic accelerates the need of alternative new effective therapy with fast response after treatment which as already is discuss above can be the photodynamic method against viruses [1]. PDI is already well accepted for decontamination of blood products, in disinfection of clinical consumables and surfaces especially in infection unit, in the catheters cleaning and the air-tube for saturation and inhalation. The future prospect for photosensitizers such as the studied in the present work may be prospective for application in the emergency cases caused by viruses. PDI and the other light-based strategies were summarized as an useful approach to reduce viruses transmission through atmosphere, water, contact places and in regular human activities [2].
- Diogo A. Mendonça, Iris Cadima-Couto, Carolina C. Buga, Zoe A. Arnaut, Fabio A. Schaberle, Luis G. Arnaut, Miguel A.R.B. Castanho, Christine Cruz-Oliveira, Repurposing anti-cancer porphyrin derivative drugs to target SARS-CoV-2 envelope, Biomedicine & Pharmacotherapy, Volume 176, 2024, 116768.
- 2. Caetano P. Sabino, Anthony R. Ball, Mauricio S. Baptista, Tianhong Dai, Michael R. Hamblin, Martha S. Ribeiro, Ana L. Santos, Fábio P. Sellera, George P. Tegos, Mark Wainwright, Light-based technologies for management of COVID-19 pandemic crisis,
Journal of Photochemistry and Photobiology B: Biology, Volume 212, 2020, 111999.
Comments 3: Check the structures in Figure 1 – one nitrogen is missing from the macrocycle (on the right in both structures). Please check bonds with hydroxy group in gallium phthalocyanine.
Response 3: We thank the esteemed reviewer for pointing out the omission. The figure has been corrected.
Comments 4: There are no units for concentration in Table 1.
Response 4: We thank the esteemed reviewer for the pointed out omission. The unit of measurement is indicated in the table.
Comments 5: Grammar should be checked and corrected throughout the manuscript (e.g. line 140 ‘…were propagate…’, line 330: ‘PDI procedure described so far has mainly be limited to the treatment’ etc.).
Response 5: Grammar was revised throughout the manuscript.